# Evaluating China Food’s Fertilizer Reduction and Efficiency Initiative Using a Double Stochastic Meta-Frontier Method

**DOI:** 10.3390/ijerph19127342

**Published:** 2022-06-15

**Authors:** Xi Chen, Mingzhe Pu, Yu Zhong

**Affiliations:** Institute of Agricultural Economics and Development, Chinese Academy of Agricultural Sciences, Beijing 100082, China; chenxi_alice@163.com (X.C.); pumingzhe@caas.cn (M.P.)

**Keywords:** agricultural sustainability, grain fertilizer use efficiency, fertilizer reduction, agricultural environmental pollution

## Abstract

Improving the efficiency of fertilizer usage is important to achieve sustainable agricultural production. As a major agricultural producer, China formally proposed a national fertilizer reduction and efficiency initiative in 2015. Using the double stochastic meta-frontier method to measure the fertilizer use efficiency of 31 provinces in mainland China from 2005 to 2019, this study evaluates the effectiveness of the said initiative on grain production. The results show that China’s initiative has achieved some success, with the average value of fertilizer use efficiency in national grain production increasing by 2.53 percentage points. However, the changes in fertilizer use efficiency show regional heterogeneity. Specifically, the fertilizer use efficiency of the main grain-producing and marketing regions has increased significantly, while that of grain-producing-and-marketing-balanced regions has declined. Further investigation shows that this phenomenon may be related to the importance attached by local governments to the initiative and the uneven distribution of related resources.

## 1. Introductions

As of 2021, China has achieved the “eighteenth consecutive bumper grain harvest” starting from 2003. However, this achievement was accompanied by problems such as a chronic overuse of agricultural fertilizers and inefficient fertilizer application [1]. Calculated based on data from the National Bureau of Statistics of China, the discounted amount of agricultural fertilizer per unit area used in China in 2015 was 446.1 kg/hm^2^, far exceeding the internationally accepted upper limit of 225 kg/hm^2^ [2,3] for safe fertilizer use. It has been proven in many studies that excessive fertilizer application will bring a series of environmental problems. It can lead to negative externalities such as heavy metal accumulation [4,5,6], water eutrophication [7], water pollution [8], and soil acidification, which seriously hinder the sustainable development of agriculture [9] and are also detrimental to human health [10]. As a major global agricultural producer, China officially proposed a fertilizer reduction and efficiency initiative in 2015 and set the goal of achieving “fertilizer use reduction and zero fertilizer growth rate” by 2020. Based on the absolute value of the total amount of agricultural fertilizer used, China has achieved remarkable results, meeting the initial reduction target. According to the China Statistical Yearbook, the amount of agriculture fertilizer used in China has declined for the first time in history since peaking at 60,226 thousand metric tons in 2015 (see Figure 1), falling to 52,507 thousand metric tons in 2020, a decrease of 12.82%.

In China, grain crops are the dominant crop category, accounting for 70% of the total crop sown area. Therefore, it is important to evaluate specifically the fertilizer reduction and efficiency performance in the grain sector in light of the target’s achievement. However, the *China Statistical Yearbook* only publishes data on the total amount of agriculture fertilizer used, which do not allow a direct assessment of the performance of grain crops. Among the data released by other official agencies, the *National Farm Product Cost–benefit Survey* published by the Price Department of China National Development and Reform Commission includes data on the amount of fertilizer used per unit area for three staple grain crops. The survey data show a small continuous upward trend in the amount of grain fertilizer used in China in recent years (see Figure 1), in contrast to the downward trend of the total amount of agriculture fertilizer used.

Given the seeming contradiction, this study argues that the changes in grain fertilizer use efficiency must be evaluated based on input–output relationships. It specifically examines whether the target for fertilizer use reduction and efficiency was met in the grain sector. Further, it investigates whether the changes in fertilizer use efficiency in grain production vary by province according to their roles in grain production (e.g., as main grain-producing, grain-marketing, or grain-producing-and-marketing-balanced regions)? Exploring these questions will help clarify the changes in fertilizer use efficiency in China after the zero-growth fertilizer application initiative was implemented, as well as the variations in fertilizer use efficiency and change characteristics in various provinces with different grain production responsibilities. It will also allow a more comprehensive understanding and evaluation of the results of the fertilizer reduction and efficiency initiative in the grain sector and highlight relevant experiences in promoting fertilizer reduction and efficiency and sustainable agricultural development.

This study uses a double stochastic meta-frontier analysis approach to measure fertilizer use efficiency in China’s 31 provinces from 2005 to 2019. Unlike previous related studies, this study contributes as follows. First, it evaluates the effectiveness of China’s fertilizer reduction and efficiency initiative in grain crop production since 2015 from a fertilizer use efficiency perspective. Second, in contrast to the assumption implied in previous studies—that all provinces have the same production frontier in terms of grain production technology—this study refers to Zhang and Zhou [11] and considers the technological heterogeneity of grain production capacity among the main grain-producing, main marketing, and grain-producing-and-marketing-balanced provinces when calculating the fertilizer use efficiency. This method enhances the informativeness of the fertilizer use efficiency measurement results. Finally, this study’s novelty lies in its data processing approach. In the absence of official data on the amount of fertilizer used for grain, this study adjusts the amount of fertilizer used for agriculture according to the relevant data from the *National Farm Product Cost–benefit Survey* and obtains fertilizer input data for grain production in each province in a scientific manner, ensuring the accuracy of the results of fertilizer use efficiency measurement at the data level.

The remainder of this paper is structured as follows. The second part provides the institutional background of this study and the relevant literature. The third part discusses the research methodology and model setting. The fourth part presents an analysis of the measurement results of fertilizer use efficiency in grain production. The fifth part presents a further discussion of the model results. The last part discusses the main conclusions and relevant policy recommendations.

## 2. Institutional Background and Literature Review

### 2.1. Institutional Background

#### 2.1.1. China’s Fertilizer Reduction and Efficiency Initiative

In 2015, China’s Ministry of Agriculture and Rural Affairs issued the Zero Growth Action Plan for Fertilizer Use by 2020 and the Implementation Opinions of the Ministry of Agriculture on Fighting the Battle against Agricultural Surface Source Pollution, formally proposing the implementation of fertilizer reduction and efficiency actions. These documents proposed that China should achieve the goal of “one control, two reductions, and three basic” in 2020. The “two reductions” refer to “reducing the use of chemical fertilizers and pesticides and implementing zero-growth action on chemical fertilizers and pesticides. For the latter, it was specified that “from 2015 to 2019, gradually control the annual growth rate of fertilizer use to within 1% and strive to achieve zero growth in fertilizer use for major crops by 2020.” Under the guidance and requirements of the central government, local provincial governments issued relevant documents and initiated programs to implement the recommendations (see Table A1). However, it can also be seen from Table A1 that the timing of the release of official documents varied from province to province. Further analysis is presented with the discussion of the empirical results.

#### 2.1.2. Division of Grain Production Responsibility

The division of grain production responsibility resulted from a combination of factors such as economic development, resource endowment, social demand, and technological progress [12]. The concepts of main grain-producing, main marketing, and grain-producing-and-marketing-balanced regions in China began with the first round of grain purchase and marketing system reform in the 1990s [13]. The 1994 Circular of the State Council on Deepening the Reform of the Grain Purchase and Marketing System mentioned “organizing the linkage of marketing between producing regions and marketing regions.” Based on grain production and consumption, six provinces—Beijing, Tianjin, Shanghai, Fujian, Guangdong, and Hainan—are classified as the main grain-marketing regions. Subsequently, the *Opinions of the State Council on Further Deepening the Reform of Grain Circulation System*, issued in 2001, added Zhejiang Province to the main grain-marketing group. In 2003, the Ministry of Finance issued the *Opinions on Reforming and Improving Certain Policies and Measures for Comprehensive Agricultural Development*, which identified 13 provinces as the main grain-producing regions. Since then, China has formally formed grain production responsibility divisions with 13 main grain-producing provinces, seven main marketing provinces, and 11 grain-producing-and-marketing-balanced provinces.

In follow-up processes, China has continued to strengthen the pattern of grain production by capitalizing on comparative advantages through policy tilting, thereby intensifying the differences in grain production capacity among main grain-producing, main marketing, and grain-producing-and-marketing-balanced areas. In 2011, the State Council issued *the National Main Functional Area Plan*, which clearly mentions that “support for the main grain-producing areas will be increased.” In terms of food price support, the following measures were implemented in the key producing provinces: minimum purchase prices for wheat and rice, corn and soybean producer subsidies, and other support policies for key grain varieties. In terms of agricultural infrastructure construction, high-standard basic farmland planning for the main grain-producing areas accounted for 70% of infrastructure construction, while non-main grain-producing areas accounted for 30%. In terms of financial support, incentive funds are issued to large grain-producing counties. In 2020, the scale of incentive funds for large grain-producing counties was 46.67 billion *RMB*. Such incentive funds have become an important source of fiscal revenue for many major grain-producing areas, strengthening the input of grain production in these areas and directly and effectively promoting grain production [14,15]. In terms of arable land protection, subsidies for arable land protection are focused on the main grain-producing areas. In terms of agricultural fertilizer utilization, the pilot soil formula testing service is available to all agricultural counties nationwide, delivered mostly in the main grain-producing areas. As such, the main grain-producing, main marketing, and grain-producing-and-marketing-balanced areas show significant heterogeneity in various aspects of grain production. Chen et al. [16] found significant differences in grain production efficiency between the main grain-producing and main marketing areas. Luo et al. [17] showed that the policy applicable to the main grain-producing areas significantly reduces the agricultural pollution caused by chemical fertilizers.

In summary, Chinese provinces with different grain production responsibilities have significant differences in their natural resource endowment, agricultural technology development level, policy support intensity, and green agricultural technology application. Their grain production technologies are significantly heterogeneous. Therefore, it is reasonable to use grain production responsibility as the basis for grouping regions in this study, as it provides the necessary prerequisites for estimation via a meta-frontier method.

### 2.2. Literature Review

#### 2.2.1. Fertilizer Use Efficiency

Numerous studies have defined the concept of fertilizer use efficiency. Dobermann et al. [18] summarized five methods for measuring fertilizer use efficiency from both agronomic and economic perspectives. Fertilizer use efficiency from an agronomic perspective is generally measured by the fertilizer uptake rate [19]. However, as fertilizer uptake rates are mostly calculated based on strict experimental conditions and do not fully reflect the actual agricultural production of farmers, many studies have measured fertilizer use efficiency based on the economics of input–output relationships [20]. Moreover, the fertilizer output rate is a common economic indicator for measuring the efficiency of fertilizer use [21], as it reflects the quantitative relationship between crop yield and fertilizer inputs. However, its limitation is that it ignores the role of other input factors in the agricultural production process. Considering the role of other factors, based on the measure of production efficiency [22], the academic community has proposed the concept of factor use efficiency, which is the ratio of the minimum feasible factor input to the actual observed input when keeping the quantity of output and other factor inputs constant [23]. In simple terms, fertilizer use efficiency can be understood as the difference between the actual level of fertilizer inputs and the optimal level of fertilizer inputs [23,24,25].

The estimation of the factor use efficiency is based on the measurement of production efficiency, which is the gap between actual production and an efficient level of production using technology [26]. With regard to determining the production frontier, the main methods used are the non-parametric method represented by data envelopment analysis (DEA) and the parametric method represented by the stochastic frontier analysis (SFA) [27]. The SFA approach’s advantage is the ability to test the model parameters and avoid the effect of uncontrolled factors on non-efficiency [28]. Its disadvantage is that an exact setting of the production function is required. The disadvantage of DEA is that the model itself cannot be tested because of the absence of parameters, and it lacks a theoretical economic base.

Existing research has focused more on water use efficiency [29,30,31,32] and pesticide use efficiency [33,34]. The number of studies on fertilizer use efficiency in China is relatively high, owing to the chronic problem of fertilizer overuse in China [35,36,37,38,39]. Huang and Jiang [35] measured the agricultural production efficiency and fertilizer use efficiency of China’s 31 provinces using a time-varying SFA model and compared the differences in production efficiency among provinces in the eastern coastal, central, and western regions.

#### 2.2.2. Production Heterogeneity

Although some studies have involved regional heterogeneity in efficiency, they have been estimated based on the same production frontier, ignoring the possible heterogeneity of production technologies between observed individuals [40]. Baráth and Fertő [41] used the SFA model to calculate the production efficiency of two groups with technological heterogeneity and found that considering the heterogeneity of production technology is important in efficiency estimation [42]. However, the efficiency result values for the two studied groups could not be directly compared because of the lack of a reference frame. To overcome this problem, Hayami [43] introduced the concept of meta-production functions, which was subsequently developed by Battese et al. [44] and O’Donnell et al. [45]. The core idea of a meta function is that all groups share a potential production frontier, but because of differences in resource endowment, policy support, and economic structure among the groups, each group uses its frontier to approach this meta production frontier [11].

Subsequently, some studies began to explore heterogeneity using meta-frontier models. Mulwa et al. [46] studied corn production efficiency in western Kenya using meta-frontier DEA and Tobit regression. Wang et al. [47] used a meta-frontier DEA approach to assess the energy efficiency of Chinese provinces. Zhong et al. [48] used a meta-frontier approach to measure the total-factor energy efficiency in 30 Chinese provinces from 1997 to 2016 based on the non-radial directional distance function of DEA and explored the reasons for regional differences in energy efficiency. These studies methodologically account for the heterogeneity of production across different groups but cannot avoid the non-parametric drawback of the DEA approach, that is, the inability to test the model parameters statistically. The mixed approach proposed by Battese et al. [44] and O’Donnel et al. [45] can partially overcome the shortcomings of the DEA approach. However, the mixed approach has two major shortcomings. First, the second step in the mixed approach uses a linear programming approach, which fails to give a meaningful statistical interpretation [49]. Second, the first step of the SFA estimation and the second step of the linear programming estimation in the mixed approach lack consistency. To overcome this problem, Huang et al. [49] proposed a double SFA method to estimate the efficiency of different technology groups, allowing the estimates to have both desirable statistical properties and statistical inference. Zhang and Zhou [11] used this method to explore the impact of different urban grouping criteria on energy efficiency. Bravo-Ureta et al. (2020) [50] analyzed the impact of canal irrigation projects on farm households in the Philippines using the stochastic meta-frontier method to measure the difference in agricultural productivity between farm households covered by canal irrigation projects and those not covered by the projects.

Synthesizing the previous literature, we found two main shortcomings in the studies that measured the fertilizer use efficiency in China. First, because of the limitation of fertilizer statistics, several studies have explored the fertilizer use efficiency of agricultural production as a whole, and few have studied the fertilizer use efficiency of grain. Second, because of the different resource endowments, agricultural policies, and technology levels among different provinces in China, especially the heterogeneity of production technologies among the provinces in the main grain-producing, main marketing, and grain-producing-and-marketing-balanced regions, there may be different production frontiers among provinces in different regions. To remedy these deficiencies, this study divides 31 Chinese provinces according to their grain production responsibilities and uses a double stochastic meta-frontier method to measure the fertilizer use efficiency in grain production in China.

## 3. Materials and Methods

### 3.1. Method and Model Setting

#### 3.1.1. Fertilizer Distance Function and Stochastic Frontier Model

Following Färe and Primont [51], we defined grain inputs and outputs to constitute a production technology set. In this study, four main input variables and one output variable, namely, *land* (*A*), *labor* (*L*), *machinery* (*M*), *fertilizer* (*F*), and *grain production* (*Y*), were selected to form the following production technology set *T*:(1)T={(A,L,M,F,Y): (A,L,M,F) can produce Y}

Malmquist [52] and Shepherd [53] introduced the concept of the distance function to describe the production technology, making it possible to measure efficiency and productivity. Referring to Zhou et al. [54], this study applied the input-oriented Shepherd’s fertilizer distance function, defined in Equation (2):(2)$$D_{F}\left(A,L,M,F,Y\right)=sup\left\{a:\;\left(A,L,M,F/a,\;Y\right)\right\}$$

From Equation (2), fertilizer use efficiency can be expressed as the ratio of potential fertilizer input to actual fertilizer used, as in Equation (3) below:(3)$$FEI=\frac{F^{*}}{F}=\frac{F/a}{F}=\frac{1}{a}=1/D_{F}\left(A,L,M,F,Y\right)$$

In general, the observation point is located inside the production frontier, and the potential fertilizer input is smaller than the actual fertilizer input, that is, *FEI* < 1, and $D_{F}\left(A,L,M,F,Y\right)>1$, where *FEI* means fertilizer efficiency and $D_{F}\left(A,L,M,F,Y\right)>1$ is the distance function. If the observation point is located on the production frontier, *FEI* = 1, reaches the maximum value. Common forms of distance function include linear function, Cobb–Douglas (C–D) function, quadratic function, trans-log function, and so on. In this study, we adopted the common form of the C–D production function for the agricultural production function, whose Shepherd fertilizer distance function can be expressed as in Equation (4): (4)$$lnD_{F}\left(A,L,M,F,Y\right)=\alpha_{0}+\alpha_{A}lnA+\alpha_{L}lnL+\alpha_{M}lnM+\alpha_{F}lnF+\alpha_{Y}lnY$$

Since the Shepherd distance function satisfies homogeneity (Zhou et al., 2012) [46], Equations (5) and (6) can be obtained as follows:(5)$$D_{F}\left(A,L,M,F,Y\right)=F\cdot D_{F}\;\left(A,L,M,1,\;Y\right)$$
(6)$$lnD_{F}\left(A,L,M,F,Y\right)=lnF+lnD_{F}\;\left(A,L,M,1,\;Y\right)$$

Bringing Equation (4) into Equation (6), we obtained Equation (7):(7)$$lnD_{F}\left(A,L,M,F,Y\right)=lnF+\alpha_{0}+\alpha_{A}lnA+\alpha_{L}lnL+\alpha_{M}lnM+\alpha_{Y}lnY$$

On this basis, we defined the random error, $v\sim N\left(0,\sigma_{v}^{2}\right)$ and the inefficient part $\mu =lnD_{F}\left(A,L,M,F,Y\right)$, and substituted them into Equation (7) to obtain Equation (8) as follows:(8)$$-lnF=\alpha_{0}+\alpha_{A}lnA+\alpha_{L}lnL+\alpha_{M}lnM+\alpha_{Y}lnY+v-\mu ,$$
where *v* and μ are uncorrelated, and *v* is a random disturbance term after separating the inefficient part of the fertilizer in the production process, which follows a normal distribution. 

#### 3.1.2. Double Stochastic Meta-frontier Model

Because there are significant discrepancies in grain production capacity and agricultural utilization efficiency among the provinces in the main grain-producing, main marketing, and grain-producing-and-marketing-balanced areas, this study used a double stochastic meta-frontier model. The first step was to estimate the within-group frontier. In this study, the 31 provinces were divided into three groups based on the classification criteria of grain production responsibilities, and these three groups have heterogeneous production technologies. The production technology of each group is redefined as in Equation (9):(9)$$T^{g}=\left\{\left(A,L,M,F,Y\right):\left(A,L,M,F\right)\;can\;produce\;\left(Y\right)\right\},\;g=1,2,3$$

The fertilizer distance function for each group is expressed as:(10)$$D_{F}^{g}\left(A,L,M,F,Y\right)=sup\left\{\beta :\;\left(A,L,M,F/\beta ,\;Y\right)\in T^{g}\right\}$$
and the group’s fertilizer use efficiency is expressed in Equation (11):(11)$$FEI^{g}=1/D_{F}^{g}\left(A,L,M,F,Y\right)$$

Similar to Equation (7), the fertilizer distance function for group *g* can be expressed as in Equation (12):(12)$$-lnF^{g}=\alpha_{0}+\alpha_{A}lnA+\alpha_{L}lnL+\alpha_{M}lnM+\alpha_{Y}lnY+v-\mu$$

The equation above was estimated using the SFA model, and the fertilizer use efficiency for each group was obtained [55]. As shown in Equation (13), there is an estimation error between the estimated and actual values of fertilizer:(13)$$lnF^{g}=ln{\hat{F}}^{g}\left(A,L,M,F,Y\right)+\tilde{v},\;\tilde{v}=\hat{v}-v$$

In the second step, we assumed that each group’s production technology $T^{g}$ belongs to the set of meta production technologies $T^{m}$, whose technology function expression is given by Equation (14):(14)$$T^{m}=\left\{T^{1}\cup T^{2}\cup T^{3}\right\}=\left\{\left(A,L,M,F,Y\right):\left(A,L,M,F\right)\;\;can\;produce\;\left(Y\right)\right\}$$

The meta-frontier fertilizer distance function can be expressed as: (15)$$D_{F}^{m}\left(A,L,M,F,Y\right)=sup\left\{\lambda :\;\left(A,L,M,F/\lambda ,\;Y\right)\in T^{m}\right\}$$

We then substituted the fertilizer input estimates for each group in the first step into the second step [56]. Since the meta-frontier included all of the groups’ stochastic frontiers [57], the value of the fertilizer distance function for the meta-frontier was smaller than the value of the fertilizer distance function for the groups (see Figure 2). *TGD* denotes the difference between the within-group frontier and the meta-frontier, as shown in Equation (16):(16)$$ln\left(D_{F}^{m}/D_{F}^{g}\right)=lnTGD$$

Based on the above, the fertilizer use efficiency of the meta-frontier is:(17)$$FEI^{m}=1/D_{F}^{m}\left(A,L,M,F,Y\right)$$

Further, the fertilizer use efficiency of the meta-frontier can be defined as in Equation (18):(18)$$FEI^{m}=\frac{1}{D_{F}^{m}\left(A,L,M,F,Y\right)}=\frac{1}{TGD*D_{F}^{g}}=\frac{1}{TGD}*FEI^{g}$$

Taking the logarithm of Equation (18), the relationship $-lnF^{g}=-lnF^{m}-lnTGD$ can be obtained. Combining this with Equation (13), the following relationship was obtained: (19)$$-ln{\hat{F}}^{g}\left(A,L,M,F,Y\right)=-lnF^{m}\;\left(A,L,M,F,\;Y\right)+v^{m}-\mu^{m}$$
where $v^{m}=-\tilde{v}$, and $\mu^{m}=lnTGD$. Given that $TGD=E\left(\exp\left(-u^{m}\right)\right)$, we could calculate *TGD*. For convenience, we defined the inverse of *TGD* as *TGR*. The fertilizer use efficiency of the meta-frontier could then be expressed as in Equation (21):(20)TGR=1/TGD
(21)$$FEI^{m}=TGR*FEI^{g}$$

According to Equation (21), the larger the *TGR*, the larger the fertilizer use efficiency value based on the meta-frontier measure. In theory, *TGR* represents the technical gap between the group frontiers and the meta-frontier in terms of fertilizer efficiency: the larger the *TGR*, the closer these frontiers are. Conversely, if the group frontiers are further away from the meta-frontier, the lower is the measured fertilizer use efficiency.

Figure 2 further demonstrates the relationship between the meta-frontier and the groups’ frontiers. The horizontal axis is the output y, and the vertical axis is fertilizer input *F*. For observation point A, when other inputs remain unchanged, its fertilizer use efficiency is $OF_{3}/OF_{2}$ relative to the group frontier, and $OF_{3}/OF_{1}$ relative to the meta-frontier. In addition, the ratio of the two efficiencies is *TGR*, and
$$TGR=\frac{OF_{3}}{OF_{2}}/\frac{OF_{3}}{OF_{1}}=\frac{OF_{3}}{OF_{2}}.$$

### 3.2. Data

#### 3.2.1. Data Sources

Based on the availability of data, this study collected provincial panel data on grain inputs and outputs for 31 provinces in mainland China from 2005 to 2019. The total grain production is the output variable, and the production input variables are inputs such as *land (A)*, *labor (L)*, *fertilizer (F),* and *agricultural machinery (M)*. In terms of the data, the main problem in measuring the grain fertilizer use efficiency is the lack of official statistics. Because the agricultural input indicators published by the National Bureau of Statistics of China concern only the quantity of agricultural production inputs, including the number of rural primary sector labor, the amount of fertilizer used in agriculture, and the total power of agricultural machinery, input data dedicated to the grain production component are lacking. Previous studies have tended to study the fertilizer use efficiency of agriculture or have been limited by the availability of data to directly examine the fertilizer use efficiency in grain production using data on agricultural inputs as a whole. Meanwhile, this study builds on previous research by adjusting official statistics to reflect the true input–output relationship in grain production in China.

First, the most important input indicator in this study—the amount of fertilizer used in grain production data—needs adjustment. The data in the China Statistical Yearbook on the amount of fertilizer applied to agriculture as a whole reflect the amount of fertilizer input to the agricultural industry, which needs to be adjusted to the amount of fertilizer input into grain production. To calculate the amount of fertilizer applied to grain production, a prerequisite assumption must be made. We need to assume that the proportion of fertilizer applied per unit area for each crop variety in each province is the same as the proportion of fertilizer applied per unit area for each crop variety nationwide in a particular year. There is a general ratio of the amount of fertilizer applied among different crops. The ratio of fertilizer applied per unit area for different crops in each of the provinces is not exactly the same but is relatively close to the ratio of the amount of fertilizer applied per unit area for different crops nationwide. On this basis, using the data on the amount of fertilizer used for each crop type in the *National Farm Product Cost–benefit Survey*, the sown area of each crop type in each province in the *China Statistical Yearbook*, and the data on the quantity of fertilizer used to agriculture in each province, the amount of fertilizer used in grain production in each province was calculated. The calculation is based on the following formula:(22)$$\{\begin{array}{l} \overset{p}{\underset{k=1}{\sum }}X_{it}^{k}\times A_{it}^{k}=F_{it}^{ari}\;\;\;\;\;\;\;\;\;\;\;\;\;\;\;\;\;\;\;\;\;\;\;\;\;\;\;\;\;\;\;\;\\ X_{it}^{1}:X_{it}^{2}:X_{it}^{3}:\ldots X_{it}^{p}=F_{t}^{1}:F_{t}^{2}:F_{t}^{3}:\ldots :F_{t}^{p} \end{array}$$
where $X_{it}^{k}$ denotes the fertilizer applied per unit area of crop *k* in province *i* in year *t*. $A_{it}^{k}$ denotes the sown area of crop *k* in province *i* in year *t*. $F_{it}^{ari}$ denotes the discounted amount of fertilizer applied for agriculture in province *i* in year *t*. Equation (19) indicates that the sum of the amounts of fertilizer used for each type of crop is the fertilizer use amount for agriculture. 

$F_{t}^{k}$ denotes the national average fertilizer used per unit area for crop *k* in year *t*. Equation (22) indicates that the proportion of fertilizer application for each crop type in each province in year *t* is the same as the proportion of the national fertilizer application for each crop type in year *t*. The data for $A_{it}^{k}$ and $F_{it}^{ari}$ are taken from the *China Statistical Yearbook*, and the data for $F_{t}^{k}$ are taken from the *National Farm Product Cost–benefit Survey*.

The second issue concerns the data on agricultural machinery inputs. In previous relevant studies, the total power of agricultural machinery in the *China Statistical Yearbook* was used for estimation. However, given that the proportion of grain production in the overall agricultural economic activities varies from province to province, this indicator cannot accurately reflect the machinery inputs in grain production. To address this issue, this study adds up the machine sown area of rice, maize, wheat, and soybean in each province, as provided in the *China Agricultural Machinery Industry Yearbook,* to obtain the machine sown area for the grain data, and calculates the machine sown ratio for grains, so as to represent the input of agricultural machinery in grain production. Given that the *China Agricultural Machinery Industry Yearbook* started to include agricultural mechanization operations in each province in 2005, and the latest data are up to 2019, this study uses provincial panel data from 2005 to 2019 for estimation.

Some shortcomings remain in the data that are difficult to address because of statistical limitations. For the labor input data, the amount of labor engaged in grain production cannot be used. This study refers to previous relevant literature and uses the number of people employed in the primary rural sector as a proxy, which is derived from the *China Rural Statistical Yearbook*.

In summary, most of the data in this study come from the *China Statistical Yearbook*, the *China Agricultural Machinery Industry Yearbook,* and the *China Rural Statistical Yearbook*, with some of the data that need to be supplemented coming from the *60 Years of New China Statistical Compendium* and the statistical yearbooks of each province.

#### 3.2.2. Data Descriptive Statistic

Table 1 shows the descriptive statistics of grain production data. Codes 1, 2, and 3 in column 3 represent the main grain-producing, main marketing, and grain-producing-and-marketing-balanced areas, respectively. Table 1 shows significant differences in grain production among the different groups, confirming the existence of production heterogeneity among these groups. Therefore, this study adopts the stochastic meta-frontier model to evaluate the fertilizer use efficiency of grain production in China.

## 4. Results 

### 4.1. Grain Fertilizer Use Efficiency Calculation Results

The 31 provinces are divided into three groups according to the classification criteria of grain production responsibilities, and the production frontier of each group and the meta-frontier of the three groups are estimated separately. Table 2 shows the results of the estimated coefficients of the stochastic meta-frontier and the respective functions of the three groups. The results show that most of the estimated coefficients are significant at the 10% statistical level. Based on Equations (14)–(18) provided in the preceding section, the grain fertilizer use efficiency of China’s 31 provinces from 2005 to 2019 was calculated.

Figure 3 shows the changes in the average value of fertilizer use efficiency for grain production in China’s 31 provinces from 2005 to 2019. In terms of national averages, grain fertilizer use efficiency has been fluctuating and declining until 2016, after which grain fertilizer use efficiency has taken an upward trend. The timing of this turnaround roughly coincides with China’s fertilizer reduction and efficiency initiative, whose implementation began after 2015. This finding indicates that from a national perspective, this initiative has achieved significant results in the area of grain production as a whole.

Figure 4 shows the trends in the average grain fertilizer use efficiency of the main grain-producing, main marketing, and grain-producing-and-marketing-balanced provinces from 2005 to 2019. Figure 4 demonstrates that, first, significant differences exist in the fertilizer use efficiency levels of the three groups. Specifically, the mean value of fertilizer use efficiency in provinces belonging to the main grain-producing group is higher, fluctuating between 0.9 and 1. The provinces belonging to the grain-producing-and-marketing-balanced group have the second-highest fertilizer use efficiency, fluctuating between 0.85 and 0.9 in recent years. The provinces belonging to the main grain-marketing group have the lowest mean value of fertilizer use efficiency and show a more pronounced difference compared with the first two groups, fluctuating between 0.7 and 0.8. 

These results are consistent with the principle of zoning responsibility for grain production. Because the main grain-producing areas have the advantages of scale operation and production technology, the degree of specialization and organization of grain production in these areas is higher [58], and the ability to allocate production factors comprehensively is higher. In addition, given that the main grain-producing and grain-producing-and-marketing-balanced areas bear the production burden of national food security, central financial inputs and agricultural policies are also biased in their favor, further promoting the main production areas to enhance green production methods and improve the efficiency of fertilizer use. The study by Luo et al. [17] has demonstrated that the policy applied to the main grain-producing areas reduces the use of chemical fertilizers. Second, significant differences are observed in the trends of fertilizer use efficiency among the three groups. The average value of fertilizer use efficiency in the main grain-producing and main marketing areas shows an increasing trend after 2016, while the average value in the grain-producing-and-marketing-balanced areas shows a decreasing trend. This finding reflects the differences in the implementation of fertilizer reduction and efficiency actions in different provinces with different grain production responsibilities.

Table 3 further compares the changes in fertilizer use efficiency in China’s 31 provinces after the fertilizer reduction and efficiency initiative started in 2015. Table 3 shows that among the 13 major grain-producing provinces, fertilizer use efficiency increased in 10 provinces, but not for Liaoning, Jilin, and Inner Mongolia, where fertilizer use efficiency decreased slightly. Six provinces: Hunan, Jiangxi, Hubei, Henan, Shandong, and Anhui, saw an increase of more than 5%. Notably, the four northeastern provinces (Heilongjiang, Liaoning, Jilin, and Inner Mongolia), as the base of China’s commodity grain production, are the bottom four among the 13 main grain-producing provinces in terms of fertilizer use efficiency growth. Moreover, the fertilizer use efficiency of three provinces, namely, Liaoning, Jilin, and Inner Mongolia, also decreased slightly from 2015 to 2019. In the seven main marketing provinces, in addition to Beijing, fertilizer use efficiency declined, the remaining six provinces have improved, and Tianjin, Shanghai, Zhejiang, Guangdong, and Fujian provinces had a growth rate of more than 10%. Among the 11 provinces in the grain-producing-and-marketing-balanced area, fertilizer use efficiency decreased in all 10 provinces, except in Xinjiang, which experienced a slight increase.

### 4.2. The Fertilizer Use Efficiency of Various Grain Crops

To further explore the changes in the fertilizer use efficiency in the production of different grain types, this section measured the fertilizer use efficiency in the production of the four staple grain crops in China: corn, wheat, rice, and soybean, using data from the *National Farm Product Cost–benefit Survey*. As can be seen from Figure 5 and Figure A2, the absolute value of fertilizer use efficiency in soybean production is the lowest among the four staple grain crops, while those of wheat, corn, and rice are relatively high and close to each other. These results indicate that the current fertilizer input utilization in the production of the three major staple cereals of maize, wheat, and rice in China is relatively close to the potential optimal input. Compared with the production of cereals, there is more room to improve the national fertilizer use efficiency in soybean production. Moreover, substantial differences are observed in soybean production technology and fertilizer use levels among the 13 major soybean-producing provinces. For example, the fertilizer use efficiency of soybean production in provinces such as Liaoning, Heilongjiang, Yunnan, and Shandong has reached more than 0.9, while those of Anhui and Chongqing are below 0.55, with large gaps among the sample provinces (see Figure A1). Furthermore, the fertilizer use efficiency for all four major grain crops shows a small fluctuating upward trend after 2016 (see Figure 5), and the effectiveness of China’s fertilizer reduction and efficiency actions since 2015 is also evidenced by the data from the *National Farm Product Cost–benefit Survey*. 

## 5. Discussion

According to the above, the changes in fertilizer use efficiency show obvious regional heterogeneity. This study clarifies the results from two perspectives: the implementation of fertilizer reduction and efficiency actions and the policy of assigning grain production responsibilities. As far as the main grain-producing provinces are concerned, agricultural production is a key responsibility. The main grain-producing provinces are undoubtedly the major force in fertilizer reduction and efficiency actions, and they have a greater responsibility and incentive to promote green agricultural production and implement fertilizer reduction and efficiency actions. For example, Hunan Province, which has seen the largest increase in fertilizer use efficiency, introduced the *Hunan Province to 2020 Zero Growth in Crop Fertilizer Use Action Implementation Plan* as early as 2015, setting a target of negative growth in fertilizer use for the province as a whole by 2020. In the implementation plan, key tasks and demonstration tasks were divided and distributed to every municipality to promote the coordination of fertilizer reduction and efficiency work at the provincial, municipal, and county levels. Shandong Province has also set up a technical expert steering group for fertilizer reduction and efficiency actions to strengthen technical guidance services for fertilizer reduction and efficiency. 

Second, as far as resource allocation is concerned, policies and projects such as soil formula fertilization projects are also mostly tilted toward the main grain-producing areas. In addition to green agriculture, organic agriculture is relatively developed as well, with the policy tilted further to promote the main grain-producing provinces to improve their fertilizer use efficiency gradually. However, for the commercial grain production bases in the four northeastern provinces, whose cultivation structure is dominated by grain crops and which bear the brunt of China’s grain production, ensuring increased and stable grain production is the first objective and main prerequisite, not the goal of chemical fertilizer reduction and efficiency increase. If a conflict arises between fertilizer reduction and yield increase, the former must give way to the latter. In addition, based on the information presented in Table A1, none of the four provinces has issued and published targeted action guidelines and fertilizer reduction and efficiency action targets in the first place. This may lead to local governments and agricultural departments not paying enough attention to fertilizer reduction and efficiency in the production process and lacking the motivation to implement it actively.

In the case of the main grain-marketing areas, their fertilizer use efficiency is relatively the lowest in absolute terms. This means that their potential for efficiency improvement is the greatest. Compared with the other two groups, the main grain-marketing areas have relatively the least responsibility for grain production and the weakest agricultural production base. Nevertheless, they have the advantage of being economically developed and having sufficient financial resources to promote green production technologies that are conducive to fertilizer reduction and efficiency through subsidies and other means of promoting fertilizer reduction and efficiency actions. In addition, economically developed areas usually have higher environmental requirements and exert greater efforts to remediate sources of agricultural surface pollution such as fertilizers, and they often have more incentive to implement fertilizer reduction and efficiency targets. For example, Shanghai has arranged municipal financial funds for the issuance of agricultural green production subsidies, which include funds for the promotion of production technologies such as the application of organic fertilizers, soil testing and formula fertilizers, water and fertilizer integration, as well as funds for the resource utilization of agricultural waste. Moreover, the provinces belonging to the main grain-marketing group attach more importance to the promotion of chemical fertilizer reduction and efficiency actions, and it can be seen from the relevant documents listed in Table A1 that the provinces in the main grain-marketing group generally started earlier with fertilizer reduction actions. In particular, all of Tianjin, Shanghai, and Zhejiang, where fertilizer use efficiency has increased rapidly, responded to the central government’s call in 2015 to formulate long-term implementation plans for zero growth in fertilizer use by 2020. Shanghai also quantified its target and issued it to all districts, which has regulated Shanghai’s need to reduce the city’s average fertilizer use by 20%, from 29.5 kg/mu in 2015 to 24 kg/mu by 2020.

In the case of grain-producing-and-marketing-balanced regions, their responsibility for grain production is relatively limited compared with that of the main grain-producing provinces. As such, they have less incentive to implement fertilizer reduction targets. Compared with the other two regions, the grain-producing-and-marketing-balanced areas lack the corresponding resource support, both central policy and central funding support, as well as financial strength for back-up purposes. In addition, an analysis of Table A1 reveals that provincial governments in grain-producing-and-marketing-balanced areas generally introduced specific programs on fertilizer reduction and efficiency actions later than the regions from the other two groups and lacked the setting and requirements for action targets, which may have led to minimal attention in the implementation process at the lower levels.

Based on the above analysis and discussion, we draw three policy implications. First, local governments should be encouraged and urged to introduce and improve targets and plans for fertilizer reduction and efficiency actions. One way to do this is to set long-term fertilizer reduction and efficiency targets. Provincial governments could set milestones according to their own situation to encourage lower-level governments to take action to strengthen the implementation. Another is to include sustainable agricultural development in the assessment of a governor’s responsibilities for food security and to evaluate fertilizer and pesticide reduction and efficiency increase as assessment indicators for each province. Another way is to encourage each local government to implement fertilizer reduction and efficiency increase actions through the pressure of assessment. 

Second, the areas and grain types in which fertilizer reduction and efficiency are weak should be the focus of support. The central government should support areas with poor foundations and difficulties, namely, the grain-producing-and-marketing-balanced areas and the four northeastern provinces (Jilin, Liaoning, Heilongjiang, and Inner Mongolia). For the grain-producing-and-marketing-balanced areas, there is a lack of motivation and financial resources to promote fertilizer reduction and efficiency actions. The Ministry of Agriculture and Rural Development should consider this situation when setting up the relevant pilot projects and pay attention to these areas. For the northeastern provinces, local governments and scientific research groups should work together to focus on balancing the relationship between grain production and fertilizer reduction. In addition, special attention should be paid to soybean production, which has low fertilizer use efficiency. The agricultural department should clarify the special characteristics of soybean and the reasons behind the low fertilizer utilization efficiency in soybean production and further improve the fertilizer use efficiency in soybean production.

Finally, if the main grain-marketing regions and grain-producing-and-marketing-balanced regions further release the potential of fertilizer use efficiency, China’s grain fertilizer use continues to reduce and efficiency further increases, which will not only reduce the economic cost of fertilizer for agricultural cultivation but also reduce the environmental problems such as soil pollution and water pollution brought about by excessive fertilizer. In addition, as the raw materials for fertilizer processing are coal and natural gas, it will also reduce the consumption of non-renewable resources and reduce the pollution generated by the processing process.

## 6. Conclusions

China has been implementing fertilizer reduction and efficiency initiatives since 2015; however, the effectiveness of the reduction in the grain sector lacks statistical support, and further evaluation of changes in fertilizer use efficiency is needed. Previous studies on grain fertilizer use efficiency have often ignored the heterogeneity of grain production technologies arising from the differences in regional disposable agricultural resources. In this study, 31 provinces in mainland China are grouped according to the classification criteria of main grain-producing, main marketing, and grain-producing-and-marketing-balanced areas to consider the impact of the heterogeneity of production technologies on fertilizer use efficiency in provinces with different grain production responsibilities comprehensively. This study uses a double stochastic meta-frontier model based on Shepherd’s fertilizer distance function to estimate the fertilizer use efficiency. 

The key empirical results of this study are as follows: 

(1)China’s fertilizer use efficiency has been declining continuously since 2000 and started to improve once it turned around in 2016. Nationwide, the average value of fertilizer use efficiency has increased by 2.53 percentage points. This finding indicates that China’s fertilizer reduction and efficiency initiative has been effective in the grain sector and has significantly improved the fertilizer use efficiency in this sector; (2)Because of the combined effects of natural resource endowment, production experience accumulation, and financial support inclination, fertilizer use efficiency in grain production has obvious regional heterogeneity. Specifically, it is the highest in the main grain-producing areas, second highest in the grain-producing-and-marketing-balanced areas, and lowest in the main grain-marketing areas; (3)The trend of fertilizer use efficiency after 2016 also shows regional heterogeneity due to differences in the degree of government attention and the actual implementation of actions. Fertilizer use efficiency in the main grain-producing and main grain-marketing areas generally increased after 2016, but the fertilizer use efficiency in the grain-producing-and-marketing- balanced areas decreased. The central policy resource tilts toward the main grain-producing areas, and the main grain-marketing areas’ financial advantages have played an important role in promoting the implementation of fertilizer reduction and efficiency actions. The importance of the local government’s attention to the task and of the setting of reduction targets are also emphasized; (4)In terms of grain types, compared with wheat, corn, and rice production, soybean production has relatively low fertilizer use efficiency and thus, needs to be improved in the future.

There are several limitations of this study that need to be explained. First, there are accuracy issues with respect to the data. Although the study makes every effort to ensure that the empirical model data use input data of grain production, the data on the amount of labor input for grain production are substituted by the number of people employed in the primary rural industry due to the lack of official statistics. Additionally, the calculation of the amount of fertilizer inputs for grain production must rely on the assumption that the proportion of fertilizer applied per unit area for each crop variety in each province is the same as the proportion of fertilizer applied per unit area for each crop variety nationwide in a particular year. Secondly, in Section 4.2, when measuring the fertilizer use efficiency of different grain varieties using data from the *National Farm Product Cost–benefit Survey*, the fertilizer use efficiency of different varieties of crops cannot be compared directly because the production technology of different varieties of crops does not obey the same production frontier. The results can only be used as a reference to show that the fertilizer use efficiency of soybean has significant differences in the sample provinces and has room for improvement. Finally, to fully evaluate the impact of fertilizer reduction and efficiency initiatives in China, the quantitative relationship between fertilizer reduction, the environment, and human health needs to be further explored. Related studies require further contributions from natural science researchers.

## Figures and Tables

**Figure 1 ijerph-19-07342-f001:**
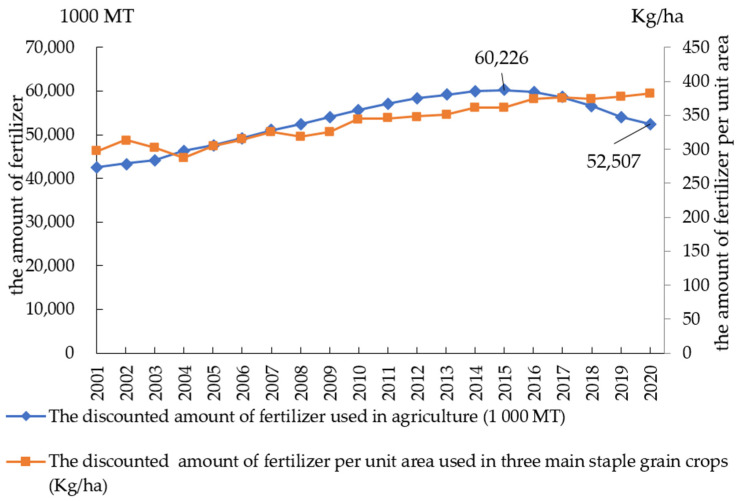
The total amount of fertilizer used in agriculture and that used in the three main staple grain crops.

**Figure 2 ijerph-19-07342-f002:**
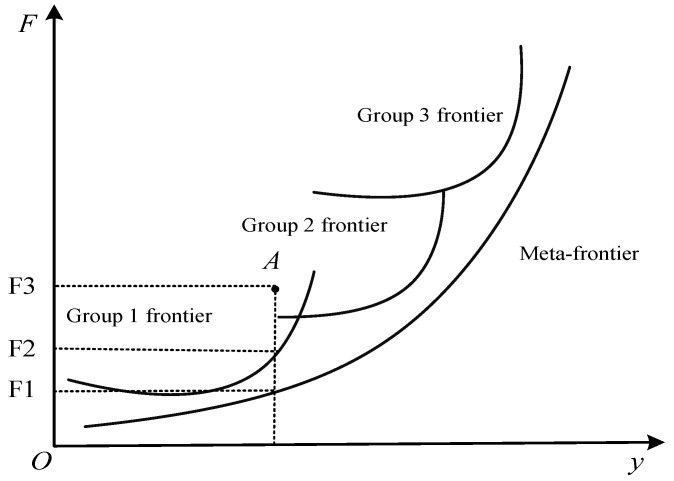
Illustration of the relationship between the meta-frontier and the groups’ frontiers.

**Figure 3 ijerph-19-07342-f003:**
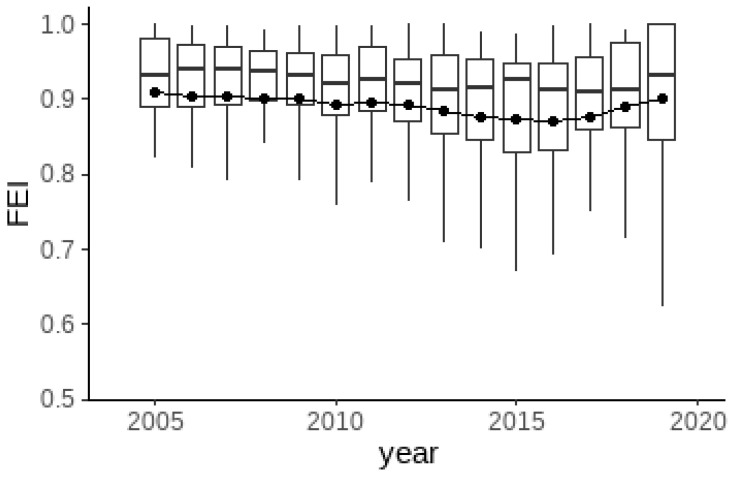
Trends in fertilizer use efficiency for grain production in China.

**Figure 4 ijerph-19-07342-f004:**
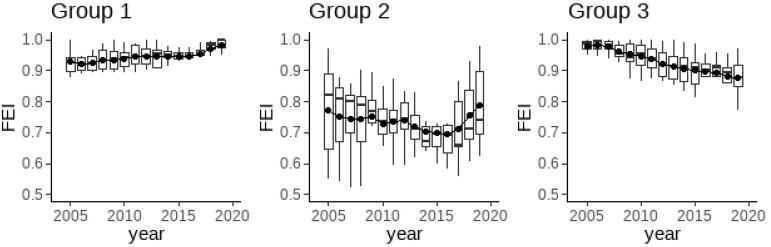
Changes in fertilizer use efficiency for regions with different food production responsibilities. Group 1, 2 and 3 represent the main grain-producing, main grain-marketing, and grain-producing-and-marketing-balanced areas, respectively.

**Figure 5 ijerph-19-07342-f005:**
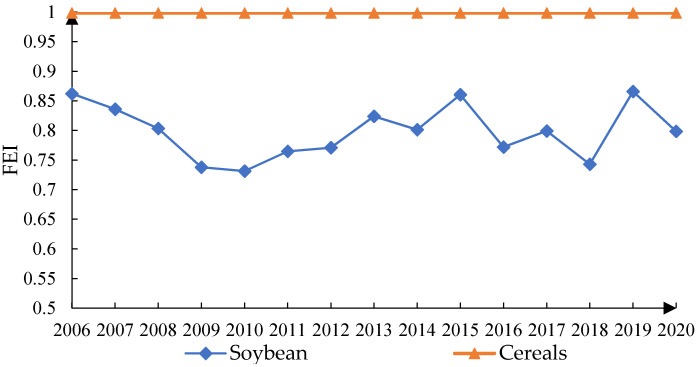
Trends in fertilizer use efficiency in the production of different grain types. The cereals FEI is the average of FEI for three crops: corn, rice, and wheat. The FEI levels of wheat, corn, and rice are relatively close and trend in the same direction, making them difficult to distinguish in the figure. Therefore, the cereal FEI is used in the figure to replace the three crop FEIs.

**Table 1 ijerph-19-07342-t001:** Descriptive statistic of grain production data.

Variables	Unit	Group	N	Mean	Std.	Min	Max
Grain production	Metric ton 10,000	All	465	1913	1680	28.76	7616
		1	195	3516	1372	1613	7616
		2	105	76.69	76.08	6.170	261.0
		3	165	115.0	81.99	4.210	263.8
Grain fertilizer input	Metric ton 10,000	All	465	179.3	143.9	4.210	716.1
		1	195	289.0	139.3	115.6	716.1
		2	105	76.69	76.08	6.170	261.0
		3	165	115.0	81.99	4.210	263.8
Grain sown area	Hectare 1000	All	465	3645	3054	46.52	14,338
		1	195	6397	2627	3046	14,338
		2	105	778.5	739.7	46.52	2787
		3	165	2216	1250	169.4	4277
Labor input	Person 10,000	All	463	910.3	712.8	37.09	4276
		1	194	1357	726.1	466.2	4276
		2	105	384.9	424.0	37.09	1610
		3	164	718.9	499.5	86.39	1709
Grain machinery sowing area	Hectare 1000	All	465	1969	2827	0.0600	14,053
		1	195	4042	3327	21.95	14,053
		2	105	135.8	124.3	0.0600	383.0
		3	165	684.9	768.1	0.220	2538

Notes: Codes 1, 2, and 3 in the group represent the main grain-producing, main grain-marketing, and grain-producing-and-marketing- balanced areas, respectively.

**Table 2 ijerph-19-07342-t002:** Estimation results of the meta-frontier and the groups’ frontier.

Variables	(1)-*Fertilizer*	(2)-*Fertilizer*	(3)-*Fertilizer*	(4)-*Fertilizer*
*Fertilizer Inputs*	Meta-Frontier	Group 1	Group 2	Group 3
Grain production	−0.470 ***	−0.381 ***	0.851 ***	−0.683 ***
Labor input	−0.185 ***	−0.216 ***	−0.381 ***	0.163 **
Grain sown area	−0.182 ***	−0.616 ***	−1.158 ***	−0.326 **
The ratio of grain machine sowing	−0.194 ***	0.109 *	−0.087 ***	−0.246 **
Constant	−23.114 ***	−5.587 ***	−6.895 ***	−4.749 ***
N	463	194	105	164

Notes: Robust t-statistics are in parentheses. ***, **, and * denote significance at the 1%, 5%, and 10% levels, respectively. The dependent and independent variables, except the ratio of grain machine sowing, are in logarithmic form.

**Table 3 ijerph-19-07342-t003:** Ranking of fertilizer use efficiency in 31 provinces.

Region	Province	2015	2019	Change (%)
Group 1	Hunan	0.929003	0.999305	7.57
	Jiangxi	0.932921	0.999312	7.12
	Hubei	0.938174	0.999318	6.51
	Henan	0.940219	0.999309	6.29
	Shandong	0.944247	0.99932	5.84
	Anhui	0.950215	0.999313	5.17
	Hebei	0.960072	0.999314	4.08
	Jiangsu	0.968025	0.999313	3.23
	Sichuan	0.969519	0.999311	3.07
	Heilongjiang	0.893594	0.90473	1.24
	Liaoning	0.974804	0.97263	−0.23
	Jilin	0.954765	0.949755	−0.52
	Inner Mongolia	0.942181	0.931941	−1.08
Group 2	Xinjiang	0.835082	0.866486	3.76
	Shanxi	0.928121	0.926716	−0.15
	Shaanxi	0.910031	0.90634	−0.41
	Ningxia	0.987279	0.969887	−1.76
	Gansu	0.909234	0.891069	−2.00
	Tibet	0.969127	0.948581	−2.12
	Qinghai	0.881894	0.849597	−3.66
	Guizhou	0.811796	0.770863	−5.04
	Chongqing	0.82373	0.780809	−5.21
	Yunnan	0.925007	0.878339	−5.05
	Guangxi	0.922123	0.844835	−8.38
Group 3	Tianjin	0.694821	0.967086	39.19
	Shanghai	0.866447	0.980368	13.15
	Zhejiang	0.728888	0.82981	13.84
	Guangdong	0.669889	0.741222	10.64
	Fujian	0.643094	0.709014	10.25
	Hainan	0.597748	0.622631	4.17
	Beijing	0.708392	0.678188	−4.26

Notes: Group 1, 2, and 3 represent the main grain-producing, main grain-marketing, and grain-producing-and-marketing- balanced areas, respectively.

## Data Availability

Publicly available datasets were analyzed in this study. This data can be found here: https://data.stats.gov.cn/easyquery.htm?cn=E0103.

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
