# Peer review of "Evaluating China Food’s Fertilizer Reduction and Efficiency Initiative Using a Double Stochastic Meta-Frontier Method"

_ijerph, 2022, doi:10.3390/ijerph19127342_

Round 1

Reviewer 1 Report

# Comments to Authors:

Interesting work. The authors tried to explore the fertilizer reduction and efficiency use condition in China by using a double stochastic meta-frontier method. Three main findings were achieved according to the relevant results: (1) China's fertilizer use efficiency has been declining continuously since 2000 and started to improve once it turned around in 2016. (2) fertilizer use efficiency in grain production has obvious regional heterogeneity, and (3) the trend of fertilizer use efficiency after 2016 also shows regional heterogeneity due to differences in the degree of government attention and the actual implementation of actions. I have some questions on the data treatment and hope that all the tables and figures could be improved:

# Comment 1:

Figure 1: I am confused here, in some years (e.g. 2001-2003, 2008-2016), why the amount of fertilizer in agriculature is higher than the amount of fertilizer in the three main staple grain crops? Dose the fertilizer in agriculagure conculde that in the three main grain?

# Comment 2:

For all the Figures (Figure 1-Figure 5): What does the Y-axis number represent? Please indicate next to the Y-axis.

# Comment 3:

Table 3 and line 461-466: I am confused to the Table 3, especially the calculation method of the "Change" in Table 3. For example, for the first raw of the table, 0.999305-0.929003=0.070302. Why you use "7.03" to indicate the change percentage? this is strange. Please explain. It will be better when you add the unit of each indicator in the header raw.

# Comment 4:

Line 466: similar questions as in # Comment 3. “7.03” is not a percentage change point. I understand your means, but please add the unit of your data in all your tables and figures, to avoid some misunderstanding and confusion.

# Comment 5:

Line 551-557: Please add a new Table containing these information.

Author Response

请参阅附件。谢谢。

Reviewer 2 Report

Dear Authors,

The subject of the study is interesting and topical, with high scientific and practical importance.

Some suggestions and corrections were made in the article.

The following aspects are brought to the attention of the authors.

1.

It is recommended to follow the chapter structure of the article in accordance with Instructions for Authors, and Microsoft Word template, International Journal of Environmental Research and Public Health.

  1. Introductions
  2. Materials and Methods
  3. Results
  4. Discussion
  5. Conclusions

After reviewing and correction the content of the article on the structure of chapters, it is recommended to check the bibliographic sources in order to be in accordance with the References chapter.

2.

Equations

It is recommended to check the type of parentheses in the equations

Eg

Page 6, Equation (1)

The parentheses "( )" and "{ }" are written. "{ }" parenthesis required, or "[ ]" may be used?

3.

It is recommended that the text settings be in accordance with the Instructions for Authors, and the Microsoft Word template, International Journal of Environmental Research and Public Health.

Styles: MDPI_3.1_text

Eg.

Page 7

It is recommended to check and correct, as appropriate.

4.

Figure presentation

It is recommended that the presentation of a figure be made after the first reference to it in the text.

Eg

Figure 3, page 10

It is recommended that the presentation of a Table be made after the first reference to it in the text.

Eg.

Table 3, page 12

5.

Conclusions

The conclusions are too broad.

Their revision and synthesis is recommended, in accordance with the Instructions for Authors and Microsoft Word template, International Journal of Environmental Research and Public Health.

6.

References

The entire References chapter needs to be revised in accordance with the Instructions for Authors and Microsoft Word template, International Journal of Environmental Research and Public Health.

“Author 1, A.B.; Author 2, C.D. Title of the article. Abbreviated Journal Name Year, Volume, page range.”

Abbreviated Journal Name

Eg

Page 19, row 687

"Nutr. Cucling Agroecosyst."

Instead of

Nutr. Cycling in Agroecosyst.

Page 20, row 745

Int. J. Oper. Res.”

Instead of

Int. J. Operational Research

Volume

Page 19, row 666

4” instead of “4”

Some bibliographic sources presented the doi number, and others did not.

It is recommended to check each bibliographic source and to correct, as appropriate.

Reviewer 3 Report

This paper reports on trends of fertilizer use in China from 2005 to 2019. The idea is to avoid overuse of fertilizers, low fertilizer-use efficiency, and negative externalities but none of these was well dquantified. 

China implemented an initiative on fertilizer reduction and zero fertilizer growth rate starting in 2015 and ending in 2020. Overuse is not defined explicitly but is derived from a Chinese paper [1]. We deduce from Figure 1 that 52.507 kg/mu, i.e. 788 kg/ha of fertilizer (one mu = 0.666.5 square meters but mu is not an SI unit), is considered excessive but is it the case? This must depend on crop nutrient requirements, soil fertility levels, and the crop response models being used, not on general statistics on fertilizer sales. Moreover, it is not specified what fertilizers should be reduced in priority, given that nitrogen, phosphorus, and potassium have very different effects on crop yield, human health, and the environment. 

Fertilizer overuse in China is thought to impact on human health and the environment, but there is no data showing such relationships. I see this paper as a report on fertilizer sales in 31 regions of China grouped into three main categories (even then not defined explicitly). Each region is a combination of factors such as economic development, resource endowment, social demand, and technological progress, but what about health and environmental problems? Surprisingly, the magnitude of human health problems and environmental damages have not been documented as key factors for the grouping.  

I understand that the fertilizer growth rate must be abated down to zero growth rate between 2015 and 2020. If the objective is attained, there would be a reduction of 12.82%, distributed unevenly among regions. A double stochastic meta-frontier model was used to address China’s initiative by region. What are the starting points for the many aspects of human health and the environment in every region? Are there regional objective of reducing human health problems and environmental damage? If fertilizer applications are reduced by 12.82% (hence by approximately 100 kg/ha), what would be the nutrient source prioritized and the expected impact on human health and the environment? 

I found the text ill-organized. Normally, the following sections are strictly followed: Introduction, Materials and Methods, Results, Discussion, Conclusion. Pieces of each can be found in sections of the paper. There should be scientific hypotheses, objectives and methodology related to human health and the environment in order to enable relating fertilizer ‘overuse’ to health and environmental problems. In l. 38-39, the authors write: ‘it is important to evaluate specifically the fertilizer reduction and efficiency performance in the grain sector in light of the target’s achievement.’. The target is a policy as shown in l. 49-51, not human health and the environment on which no data have been collected. This is why I think that this paper should be submitted to a Journal in Social Sciences reporting on the success of some policy. Or simply as a progress report to the political authorities.

Round 2

Reviewer 3 Report

This paper was written by economists, focusing on the economic aspect and speculating on environmental and health issues.  However, to determine the specific quantitative relationship between fertilizer overdose, the environment and human health requires the design of rigorous biological experiments, which is not an area of expertise in agricultural economics, but rather a topic related to biology, nutrition and environmental science… The concept of fertilizer use efficiency is an economic concept that simply measures the difference between the current amount of fertilizer applied and the minimum amount of fertilizer applied, while maintaining the current level of output and other factor inputs.

Additional comments:

Define all acronyms just after the abstract.

l. 27-28: provide a reference for 225 kg ha-1. This number must be referenced absolutely. Also provide a reference for 446.1 kg/ha. Such numbers cannot be left unattended, being central issues in the paper.

l. 28: what do you mean by ‘fertilizer use intensity’? Do you mean fertilizer dosage?

l. 28: How do you define 225 kg ha-1? Do you mean N+P+K, N+P2O5+K2O or total weight of ingredients (e.g., urea+diammonium phuspahte+potassium chloride)? This is important to avoid confusion and misinterpretations of that number.

l. 30-33. Authors must specify if it includes chemical and organic fertilizers.What is the source of heavy metals? Provide more free-access references from China such as:

Peng Li, Yang Quinxian. 2021. Social network, environmental cognition and organic fertilizer application behavior of small farmers. E3S Web of Conferences 293, 03016. https://doi.org/10.1051/e3sconf/202129303016

Guiying Liu, Hualibn Xie. 2019. Simulation of Regulation Policies for Fertilizer and Pesticide Reduction in Arable Land Based on Farmers’ Behavior—Using Jiangxi Province as an Example. Sustainability 2019, 11, 136; doi:10.3390/su11010136.

Zhang, L.; Tan, X.; Chen, H.; Liu, Y.; Cui, Z. 2022. Effects of Agriculture and Animal Husbandry on Heavy Metal Contamination in the Aquatic Environment and Human Health in Huangshui River Basin. Water, 14, 549. https://doi.org/10.3390/ w14040549

Fig. 1: Kg/mu is not an acceptable unit. Change Kg/mu by kg/ha so that everybody will understand. Provide whiskers about points to make the figure statistically correct.

Provide hypotheses that will be accepted or rejected to support the conclusion.

Section 2.2.1: provide equations for the various views of fertilizer use efficiency. Equations can be understood in any language.

l. 201: how do you define the potential production function in your model? Such function must be site-specific (submitted to local factors).

l. 206-207: what is the metafrontier for your study?

Equations 1-14: check for proper spacings.

Figure 2: present numbers for maximum production.

3.2.1: present the methodology used by NBSC to collect the data. Most countries detail the procedures. If the reference to the NBSC methodology is in English, just include the reference. Otherwise, indicate the limitations of data collection in different provinces of China. Do all provinces provide uniformly the same quality of data or do you assume this should be the case?

l. 415 and table 2: what is that coefficient? The correlation coefficient? The R2 value of the model?

Figs. 3-4-5: provide whiskers about points of the graphic.

Table 3 presents numbers for 2015 and 2019 as differences between the current amount of fertilizer applied and the minimum amount of fertilizer applied? How do you interpret those numbers as fractions?

Fig. 5: corn, wheat, and rice cannot be distinguished. Maybe just mentioning cereals would suffice.

Th conclusion section is too long. Part of it should be moved to the Discussion section. Conclude on accepting or rejecting hypotheses and proposing remedies briefly.

l. 603: is it a significant decrease? It looks very small. It means that farmers show an elevated degree of insecurity due to the lack of knowledge on crop response to fertilization under their conditions. It is surprising that so big money is spent on fertilizers with so little knowledge to support actions.
